# The Effectiveness of Extra Virgin Olive Oil and the Traditional Brazilian Diet in Reducing the Inflammatory Profile of Individuals with Severe Obesity: A Randomized Clinical Trial

**DOI:** 10.3390/nu13114139

**Published:** 2021-11-19

**Authors:** Rafael Longhi, Annelisa Silva e Alves de Carvalho Santos, Anallely López-Yerena, Ana Paula Santos Rodrigues, Cesar de Oliveira, Erika Aparecida Silveira

**Affiliations:** 1Department of Nutrition, Federal University of Minas Gerais, Belo Horizonte 30130-100, Brazil; longhinutricao@gmail.com; 2Postgraduate Program in Health Sciences, Faculty of Medicine, Federal University of Goias, Goiânia, Rua 235 c/1ª s/n, Setor Universitário, Goiânia 74650-050, Brazil; annelisa.nut@gmail.com (A.S.e.A.d.C.S.); anapsr@gmail.com (A.P.S.R.); 3Department of Nutrition, Food Science and Gastronomy, XIA, Faculty of Pharmacy and Food Sciences, Institute of Nutrition and Food Safety (INSA-UB), University of Barcelona, 08028 Barcelona, Spain; naye.yerena@gmail.com; 4Department of Epidemiology & Public Health, Institute of Epidemiology & Health Care, London University College, London WC1E 6BT, UK; c.oliveira@ucl.ac.uk

**Keywords:** severe obesity, leukocytes, lymphocyte, monocytes, inflammation, nutritional intervention, olive oil, diet

## Abstract

We analyzed the effectiveness of two nutritional interventions alone and together, EVOO and the DieTBra, on the inflammatory profile of severely obese individuals. This study was an RCT with 149 individuals aged from 18 to 65 years, with a body mass index ≥ 35 kg/m^2^, randomized into three intervention groups: (1) 52 mL/day of EVOO (*n* = 50); (2) DieTBra (*n* = 49); and (3) DieTBra plus 52 mL/day of EVOO (DieTBra + EVOO, *n* = 50). The primary outcomes we measured were the-neutrophil-to-lymphocyte ratio (NLR) and the secondary outcomes we measured were the lymphocyte-to-monocyte ratio (LMR); leukocytes; and C reactive protein (CRP). After 12 weeks of intervention, DieTBra + EVOO significantly reduced the total leucocytes (*p* = 0.037) and LMR (*p* = 0.008). No statistically significant differences were found for the NLR in neither the intra-group and inter-group analyses, although a slight reduction was found in the DieTBra group (−0.22 ± 1.87). We observed reductions in the total leukocytes and LMR in the three groups, though without statistical difference between groups. In conclusion, nutritional intervention with DietBra + EVOO promotes a significant reduction in inflammatory biomarkers, namely leukocytes and LMR. CRP was reduced in EVOO and DieTBra groups and NLR reduced in the DieTBra group. This study was registered at ClinicalTrials.gov under NCT02463435.

## 1. Introduction

Severe obesity (body mass index, BMI ≥ 35 kg/m^2^) [1] is associated with an elevated risk of adverse health outcomes such as hemodynamic, neurohormonal, and metabolic alterations [1], increased cancer mortality rate at multiple specific sites [2], and reduced bone mass [3]. Furthermore, depression and anxiety are highly prevalent in individuals with severe obesity [4]. Their mental health and psychological well-being are also affected by abnormal eating behaviors, sleep disruption, and poor general quality of life. A BMI value of 40–45 kg/m^2^ is associated with a life expectancy reduction of 8–10 years which is comparable with the effects caused by tobacco [5]. With regard to cardiometabolic multimorbidity risk, this increases ten times in individuals with severe obesity compared to those with a healthy BMI [6]. In fact, the association between obesity and cardiovascular disease (CVD) has been established in animal and epidemiological studies [7]. The excess of adipose tissue in individuals with severe obesity has been associated with the release of inflammatory blood markers such as C-reactive protein (CRP), white blood cell count (WBC), fibrinogen (FB), leukocyte count and neutrophil-to-lymphocyte ratio (NLR) which constitute low-grade chronic inflammation, representing a potential link between obesity and metabolic disorders or systemic vascular complications [8,9,10,11,12,13,14,15,16]. Individuals with obesity and an increased concentration of serum insulin showed a higher total leukocyte count [16]. In addition, the lymphocyte-to-monocyte ratio (LMR) has also been suggested as a novel and useful indicator of metabolic syndrome [17].

Based on the aforementioned factors, inflammatory biomarkers have been used to evaluate the extra virgin olive oil (EVOO) intake effects on obesity [10,13,18]. The evidence, including meta-analyses, indicates that the consumption of EVOO attenuates inflammatory response in subjects with high cardiovascular risk or with metabolic syndrome [13,19,20]. The beneficial effects of EVOO have been attributed to the downregulation of the expression of proinflammatory genes and reduced levels of CRP such as the total plasma/serum concentration [19,20].

However, there are no published data on the neutrophil-to-lymphocyte ratio (NLR), lymphocyte-to-monocyte ratio (LMR), leukocytes, CRP, nutritional interventions with EVOO and a healthy dietary pattern in individuals with severe obesity. Most epidemiological studies have mainly focused on the Mediterranean diet’s effects on overweight or obesity. Therefore, taking into account the significant impact of this global public health problem and its adverse outcomes, the aim of this randomized clinical trial was to analyze the effectiveness of two nutritional interventions, i.e., EVOO and the traditional Brazilian diet (DieTBra) in reducing the NLR, LMR, leukocytes, and the CRP of individuals with severe obesity.

## 2. Materials and Methods

### 2.1. Study Design, Data Collection, and Ethical Aspects

This research is part of an RCT entitled “Effect of Nutritional Intervention and Olive Oil in Severe Obesity: A Randomized Controlled Trial”, known as the DieTBra Trial. This study included individuals with severe obesity (BMI ≥ 35 kg/m^2^), class II and III obesity, followed up for 12 weeks. Previous findings from this RCT have already been published and have demonstrated positive effects on bone health, body fat reduction, improvements in anxiety and depression symptoms, cardiometabolic risk factors reduction, and increases in muscle strength and functionality [3,21,22,23,24,25].

Data collection occurred at the Clinical Research Unit in partnership with the Nutrition in Severe Obesity Outpatient Clinic (ANOG) of the Clinical Hospital, Federal University of Goiás, Goiânia, Goiás, Brazil. This study was approved by the Research Ethics Committee of the Federal University of Goiás under the protocol number 747.792/2014. The investigation was conducted in accordance with the principles outlined in the Declaration of Helsinki. All subjects who participated in the study signed an informed written consent form. This Clinical Trial was registered at ClinicalTrials.gov under: NCT02463435.

### 2.2. Inclusion and Exclusion Criteria

This study included men and women with a BMI ≥ 35 kg/m^2^ and aged between 18 and 65 years. The exclusion criteria were history of bariatric surgery, nutritional treatment for weight loss in the last 2 years, current use of anti-obesity or anti-inflammatory drugs, having HIV/AIDS, as well as heart/kidney/hepatic insufficiency, chronic obstructive pulmonary disease, cancer, and pregnancy/lactation.

### 2.3. Participants, Randomization, and Study Phases

A total of 229 individuals with severe obesity were enrolled and screened for eligibility. Among these, 152 participants met the inclusion criteria and provided informed consent. There were three withdrawals from the study for health reasons and lack of time before randomization. After enrolment, participants underwent a baseline phase conducted in two stages within a week to answer a structured questionnaire and perform a health examination. At the end of the baseline phase, individuals were randomly assigned into three intervention groups.

Randomization was performed at the randomization.com website. The 149 individuals were allocated in a proportion of 1:1:1 into the following three intervention groups: (1) 52 mL/day of EVOO (*n* = 50); (2) DieTBra (*n* = 49); and (3) DieTBra plus 52 mL/day of EVOO (DieTBra + EVOO, *n* = 50).

The participants were followed up for 12 weeks through monthly visits. At the end of the follow up, the same data collection procedures at baseline were repeated. There were 16 losses during follow up: 7 in the EVOO group; 6 in the DieTBra group; and 3 in the DieTBra + EVOO group. The participants’ flow chart is shown in Figure 1.

### 2.4. Interventions

Participants assigned to the EVOO group were instructed to consume 52 mL of EVOO (acidity < 0.2%) daily and maintain their usual diet. The individuals were provided with enough EVOO for 4 weeks that was delivered at the end of each consultation with the registered dietitian. The EVOO was delivered in 13 mL sachets and participants were instructed to return the consumed and not consumed sachets in each consultation to assess their adherence to the intervention.

The DieTBra intervention consisted of the prescribing an individualized food plan based on the traditional Brazilian diet, which is a healthy dietary pattern characterized by the consumption of rice and beans in main meals (lunch and dinner), a small portion of lean meat (red meat, fish or chicken), and raw and cooked vegetables. Fruits, bread, milk, and dairy are consumed in small meals and participants were encouraged to eat fresh and/or minimally processed foods instead of ultra-processed foods [3,21,22,24,26]. A balanced food plan divided into four-to-six meals a day was prescribed. The prescription aimed to reduce participants’ initial body weight by 5–10% according to the BMI range, then a weekly weight reduction goal (0.5–1.0 kg/week) was set after the daily calorie reduction was determined (550–1100 kcal/day) [27]. The resting energy expenditure (REE) of participants was calculated using an equation developed for individuals with severe obesity [28]. The total energy expenditure was determined by multiplying REE by the activity factor, obtained through the Global Physical Activity Questionnaire [29] and the thermic effect of food [30]. The macronutrient distribution of the diet was set according to the recommendations of the US Institute of Medicine [31].

The DieTBra + EVOO group received the dietary intervention as described for the DieTBra group along with the supplementation of 52 mL of EVOO daily (four 13 mL sachets), which corresponded to an additional 468 kcal/day. To ensure an isocaloric prescription compared to the DieTBra group, the number of calories from the EVOO was discounted in the quantities of prescribed foods. As a measure of control and adherence to the consumption of olive oil, at each return, the participant should bring the packages of consumed and non-consumed sachets. Therefore, it was possible to calculate the average amount of mL/day that was consumed in each group with this intervention. Considering the total number of patients allocated to both groups who were prescribed EVOO, the average daily consumption was 39.62 ± 11.11 mL/day—39.96 ± 11.25 mL/day in the EVOO group and 39.28 ± 11.10 mL/day in the DietBra + EVOO group, with no statistical difference between the groups. In both groups, more than 70% of the participants consumed more than 30 mL/day.

### 2.5. Blinding

In nutritional interventions, blinding can be difficult and impractical considering the nature of the intervention itself [32]. The researchers responsible for implementing the intervention were not blinded. However, this study was designed to minimize information bias between participants from different groups, preventing them having contact with each other. Thus, each group had visits scheduled on different days of the week. Additionally, the sachets of EVOO were adequately prepared according to the recommendations of the Brazilian National Health Surveillance Agency for Clinical Trials to mask this intervention. The research team was also instructed not to mention to participants that the food supplement was olive oil, instead using the terms “dietary supplement enriched with bioactive compounds” or just “nutritional supplement”.

### 2.6. Variables and Outcomes

#### 2.6.1. Anthropometric Variables

BMI was calculated using weight and height (weight in kg/height in m^2^) [29]. Weight was measured using an electronic digital scale with 200 kg capacity and 100 g precision (Welmy), with the patient barefoot and wearing light clothes, without any objects in their pockets. Height was measured to the nearest 0.1 cm with a stadiometer coupled to the electronic digital scale. In this study, mean BMI was only used for participants’ characterization at baseline.

#### 2.6.2. Biochemical Analysis

Blood samples for biochemical analysis were collected after 12-h overnight fasting at baseline and at the end of the follow up. Hemogram and semi-quantitative CRP were evaluated. Hemogram was analyzed using resistivity/impedance method and semi-quantitative CRP through immunochemical agglutination reaction. The primary outcome of this study was the NLR and the secondary outcomes were LMR, leucocytes count, and CRP serum levels.

### 2.7. Quality Control

Standard operating procedures (SOPs) were developed to standardize all study stages and data collection. The research team was composed of qualified and registered dietitians, a psychologist, a pharmacist, physiotherapists, and nutrition undergraduate students who were trained regarding the study protocols and data collection instruments, as well as the approaches to minimizing losses to follow up, and the routine service. A pilot study was conducted to test the instruments and routine service protocols.

### 2.8. Statistical Analysis

The database was built using the EpiData software, with a double data entry to check consistency and guarantee the quality of the information. Data analysis was performed using Stata SE 12.0 software with statistical significance when *p*-value < 0.05. A descriptive analysis of means and standard deviations of the outcomes of interest was performed, with the comparison of values before and after follow up within each group and between groups.

The normality of the variables was tested by the Kolmogorov–Smirnov test. The analysis of variance was used for data with normal distribution and homoscedasticity of variances and the Kruskal–Wallis test for non-normal data to compare three independent groups. When comparing two means, the Student’s *t* test was used for data with normal distribution and the Wilcoxon or Mann–Whitney test for non-normal data. Delta values were calculated as the 12-week value minus the baseline value.

Changes in body weight and physical activity level were tested as potential covariates for the analysis of covariance (ANCOVA) of all outcomes. However, the criteria necessary to perform ANCOVA were not fully met. Therefore, it was not possible to carry out this analysis. In addition, statistical analyses were conducted with and without outliers identified by boxplot. In the analysis without outliers, the outliers were replaced by the mean (normal distribution) or median (non-normal distribution) of the observations to verify whether there would be any change in significance. As there was no change, the analysis with outliers was maintained.

## 3. Results

A total of 149 individuals were randomized into the three intervention groups: 50 participants in the EVOO group; 49 in the DieTBra group; and 50 in the DieTBra + EVOO group. No adverse effects from EVOO intake were reported. There were no significant differences for BMI, age, primary and secondary outcomes between intervention groups at baseline (Table 1).

The intervention with DieTBra + EVOO was effective in improving the leucocyte count (0.037) and the LMR (0.008) with statistically significant reductions. We also observed a reduction in CRP levels in both intervention groups, i.e., DieTBra and DieTBra + EVOO. However, this reduction was not significant (Table 2).

There were no significant differences in the mean values for all outcomes at the end of the follow up when all intervention groups were compared together and compared to each other (Table 3).

With regard to the delta values, all variables showed no significant differences between groups, even when outlier values were excluded from the analysis. Despite not being statistically significant, a clear reduction in leucocytes in all groups was observed: 468.84 ± 1757.39 mm^3^ (EVOO group), 297.16 ± 1523.73 mm^3^ (DieTBra group) and 469.98 ± 1501.62 mm^3^ (DieTBra + EVOO group). Similarly, the LMR showed a reduction in all groups: 1.57 ± 7.29 (EVOO group), 0.35 ± 2.44 (DieTBra group) and 0.84 ± 2.26 (DieTBra + EVOO group). Finally, CRP levels were reduced by 0.90 ± 18.27 mg/L in the EVOO group and by 1.04 ± 3.46 mg/L in the DieTBra group (Table 4).

## 4. Discussion

To the best of our knowledge, this was the first randomized controlled trial to compare the effectiveness of two nutritional interventions, i.e., DietBra and EVOO, in individuals with severe obesity as a strategy to modulate the inflammatory process in this condition. Our main findings showed that significant reductions in some inflammatory parameters (leukocytes and LMR) were observed when both interventions were combined–(DieTBra + EVOO). However, no significant differences were found in the simultaneous comparisons of the three intervention groups. We also observed a reduction in CRP levels in the combined intervention group. However, it was not significant. Our results make an important contribution to the understanding of how simple nutritional interventions can help improve inflammatory parameters in a high cardiovascular risk group of individuals. In addition to that, the regular consumption of EVOO offers an important biological protective mechanism, the modulation of the nuclear factor (erythroid-derived 2-like 2, Nrf2) pathway which stimulates the expression of antioxidants enzymes, superoxide dismutase, catalase, etc. [19]. It is important to note that Nrf2 is the hallmark of antioxidant responses, especially stress induced by obesity [33].

In this study, a significant reduction in leukocytes and LMR was found at the end of the DieTBra + EVOO follow up in individuals with severe obesity. Interestingly, lifestyle interventional studies have demonstrated that weight reduction has a beneficial effect on blood inflammatory markers in overweight/obese participants [34] and in obese children and adolescents [18]. High cumulative adherence to a Mediterranean diet, characterized by a high EVOO consumption, decreased the incidence of white blood cell count-related alterations in individuals at high cardiovascular risk [35]. In another study, it was demonstrated that specific components of the Mediterranean diet, particularly nuts and EVOO, were able to induce methylation changes in several peripheral white blood cell genes [36]. In patients with early atherosclerosis, diets rich in olive oil also achieved a significant reduction in inflammatory parameters [37]. The improvements in the inflammatory parameters may have clinical importance for individuals with severe obesity. Peripheral blood mononuclear cells, consisting of monocytes and lymphocytes representing cells of the innate and adaptive immune systems, are a promising target tissue in the field of nutrigenomics as they may reflect the effects of dietary modifications at the level of gene expression [38].

However, in our study, though the NLR and CRP blood concentrations did not show significant differences in any of the groups after the 12-week intervention period, it is important to highlight that the reductions could be clinically relevant. The NLR has been incorporated as a blood-based inflammatory biomarker as it may reflect the balance between innate (neutrophils) and adaptive (lymphocytes) immune responses [10]. The NLR increases by obesity grade and reveals a high count of circulating neutrophils, and in individuals with obesity, it might be considered an acute inflammatory response to a chronic inflammatory state [17]. Recently, lower-quality diets have been shown to be associated with a higher inflammatory status measured by the NLR in an older Spanish population [10]. With regard to CRP, a systematic review and meta-analysis of randomized controlled trials showed an important reduction in CRP (SMD: −0.11, 95% CI: −0.21, −0.01, *p* = 0.038) after dietary oleic acid supplementation in adults and a minimum intervention duration of 4 weeks [39]. Another randomized controlled trial demonstrated that participants had lower cellular and plasma concentrations of CRP at 3 and 5 years after they followed the Mediterranean diet supplemented with EVOO or nuts (significant reductions of 16%) [40]. Furthermore, a cross-sectional study also showed low CRP concentrations in subjects who consumed a higher amount of EVOO [41]. Recently, a daily consumption of polyphenol-rich EVOO (25 mL EVOO or low-polyphenol-refined olive oil (ROO) daily for 6 weeks) in subjects with at least one representative cardiovascular risk factor has been shown to lead to a significant reduction in plasmatic CRP (−0.40 ± 0.52 vs. 0.007 ± 0.42 mg/L, *p* = 0.01 for EVOO and ROO, respectively) [42].

Differences in our results could be attributed to the intervention time. In this sense, different studies have demonstrated that, depending on the length of time, interventions could be insufficient to improve systemic inflammatory biomarkers. For example, tomato juice intake during 20 days decreased serum concentrations of IL-8 and TNF-α in overweight and obese women, without changes in CRP levels [43]. Moreover, 12 weeks of intervention (weight-loss program + exercise) in patients with asthma and obesity were not sufficient to improve CRP levels [44]. On the other hand, after a six-month dietary intervention, there were significant decreases in CRP serum concentrations in dietary groups (hypo energetic diet rich in a-linolenic acid vs. hypo energetic diet low in a-linolenic acid) without inter-group differences in overweight-to-obese patients with metabolic syndrome traits [45].

A potential limitation of the present study is the lack of a wider range of inflammatory parameters and possibly the intervention window. Our results focused on hemogram and semi-quantitative CRP. However, it is important to mention that the biomarkers included in our clinical trial are relevant parameters to clinical practice. They are low cost and most public health services can use these methodologies.

The authors are also planning future analyses with a larger sample and longer intervention time to carefully explore the role of EVOO and DieTBra in individuals with severe obesity. We also plan to explore the evaluation of some important pro-inflammatory cytokines, such as interleukin-6 (IL-6), interleukin-1β, and tumor necrosis factor-α. There is a paucity of evidence on the role of the traditional Brazilian diet, and our results are therefore of great clinical relevance with regard to the inflammatory profiles of individuals with severe obesity.

## 5. Conclusions

In summary, DietBra combined with EVOO significantly improved the leukocyte count and LMR at the end of the follow-up period in individuals with severe obesity. Although NLR and CRP did not follow the same trend, our results suggest clinically positive changes. This was the first study to evaluate these parameters in individuals with severe obesity using a Brazilian traditional diet and EVOO.

These findings provide additional evidence that the Brazilian healthy diet combined with EVOO can positively influence inflammatory parameters in patients with severe obesity. However, future studies with a wider range of inflammatory parameters, larger samples, and a longer follow-up period to carefully explore the role of EVOO and DietBra are recommended.

## Figures and Tables

**Figure 1 nutrients-13-04139-f001:**
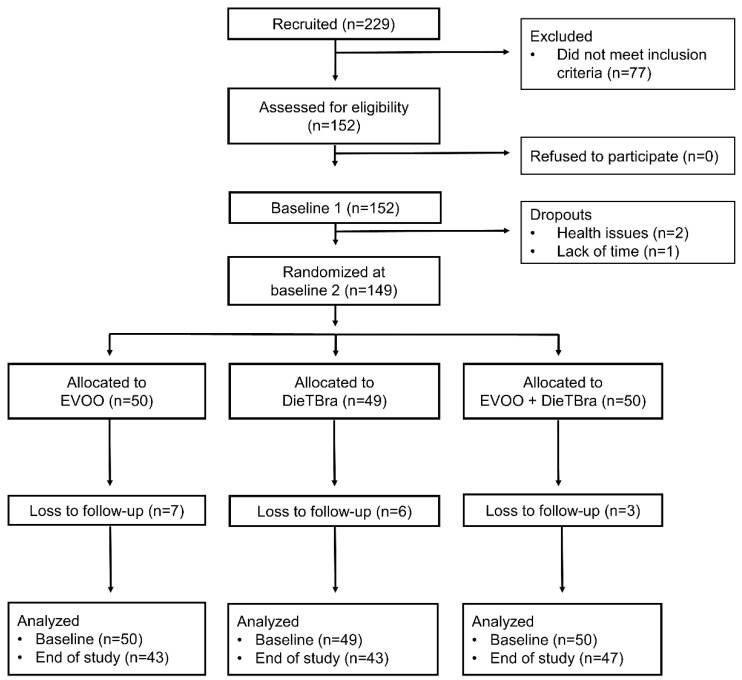
Participants’ flow chart.

**Table 1 nutrients-13-04139-t001:** Baseline characteristics according to the intervention groups in individuals with severe obesity.

	Total *n* = 149	EVOO *n* = 50	DieTBra *n* = 49	DieTBra + EVOO *n* = 50
Variables	Mean ± SD	Mean ± SD	Mean ± SD	Mean ± SD
Age, years	39.63 ± 8.82	38.14 ± 8.14	39.14 ± 8.15	41.60 ± 9.85
BMI, kg/m^2^	46.03 ± 6.40	45.77 ± 6.27	46.22 ± 6.26	46.13 ± 6.79
NLR	3.60 ± 2.88	3.44 ± 1.35	3.46 ± 1.84	3.90 ± 4.47
Leucocytes, mm^3^	8199.54 ± 2144.73	8027.30 ± 2052.14	8299.22 ± 2030.03	8274.1 ± 2366.303
LMR	5.89 ± 4.23	6.37 ± 6.64	5.65 ± 2.19	5.63 ± 2.19
CRP, mg/L	10.53 ± 7.12	10.80 ± 4.90	10.40 ± 8.62	10.45 ± 7.26
Body weight, kg	118.81 ± 19.47	117.38 ± 18.69	120.43 ± 20.85	118.64 ± 19.10
Females, *n* (%)	127 (85.23)	45 (35.43)	40 (31.50)	42 (33.07)
Males, *n* (%)	22 (14.77)	5 (22.73)	9 (40.91)	8 (36.36)

EVOO: extra virgin olive oil; DieTBra: traditional Brazilian diet; NLR: neutrophil-to-lymphocyte ratio; LMR: lymphocyte-to-monocyte ratio; CRP: C-reactive protein.

**Table 2 nutrients-13-04139-t002:** Comparison of the mean values at baseline and at the end of the follow up according to intervention groups in individuals with severe obesity.

		NLR	Leucocytes, mm^3^	LMR	CRP, mg/L	Body Weight
EVOO	Baseline (*n* = 50)	3.44 ± 1.35	8027.30 ± 2052.14	6.37 ± 6.64	10.80 ± 4.90	117.38 ± 18.69
	12 weeks (*n* = 43)	3.50 ± 1.16	7521.72 ± 2030.45	5.13 ± 1.63	13.44 ± 18.19	118.46 ± 18.93
	*P*	0.923 *	0.088 *	0.311 †	0.303 †	0.092 *
DieTBra	Baseline (*n* = 49)	3.46 ± 1.84	8299.22 ± 2030.03	5.65 ± 2.19	10.40 ± 8.62	120.43 ± 20.85
	12 weeks (*n* = 43)	3.21 ± 1.12	7981.59 ± 1930.52	5.29 ± 1.59	8.60 ± 4.07	118.85 ± 19.28
	*P*	0.513 *	0.208 *	0.365 †	0.173 †	<0.001 *
DieTBra + EVOO	Baseline (*n* = 50)	3.90 ± 4.47	8274.10 ± 2366.30	5.63 ± 2.19	10.45 ± 7.26	118.64 ± 19.10
	12 weeks (*n* = 47)	3.31 ± 1.05	7582.04 ± 1942.68	4.90 ± 1.29	8.44 ± 3.99	115.42 ± 18.68
	*P*	0.702 *	**0.037 ***	**0.008 †**	1.000 †	<0.001 *

* Student’s *t* test, paired. † Wilcoxon. Bold values indicate statistically significant results. EVOO: extra virgin olive oil; DieTBra: traditional Brazilian diet; NLR: neutrophil-to-lymphocyte ratio; LMR: lymphocyte-to-monocyte ratio; CRP: C-reactive protein.

**Table 3 nutrients-13-04139-t003:** Multiple comparisons of the mean values at the end of follow up in all intervention groups in individuals with severe obesity.

Endpoints at the End of Follow Up	EVOO (*n* = 43)	DieTBra (*n* = 43)	DieTBra + EVOO (*n* = 47)	All Groups	EVOO vs. DieTBra	EVOO vs. DieTBra + EVOO	DieTBra vs. DieTBra + EVOO
	Mean ± SD	Mean ± SD	Mean ± SD	*p*	*P*	*p*	*p*
NLR	3.50 ± 1.16	3.21 ± 1.12	3.31 ± 1.05	0.585 *	0.322 ‡	0.502 ‡	0.694 ‡
Leucocytes, mm^3^	7521.72 ± 2030.45	7981.59 ± 1930.52	7582.04 ± 1942.68	0.498 *	0.285 ‡	0.886 ‡	0.331 ‡
LMR	5.13 ± 1.63	5.29 ± 1.59	4.90 ± 1.29	0.471 *	0.651 ‡	0.463 ‡	0.207 ‡
CRP, mg/L	13.44 ± 18.19	8.60 ± 4.07	8.44 ± 3.99	0.539 †	0.387 §	0.300 §	0.853 §

* ANOVA. † Kruskal–Wallis; ‡ Student’s *t*-test, unpaired; § Mann–Whitney; EVOO: extra virgin olive oil; DieTBra: traditional Brazilian diet. NLR: neutrophil-to-lymphocyte ratio; LMR: lymphocyte-to-monocyte ratio; CRP: C-reactive protein; SD: standard deviation.

**Table 4 nutrients-13-04139-t004:** Comparison of the delta values at baseline and after the 12-week follow up.

Endpoints (12 Weeks–Baseline)	EVOO	DieTBra	DieTBra + EVOO	*p*-Value
	Mean ± SD	Mean ± SD	Mean ± SD	
∆ NLR	0.02 ± 1.08	−0.22 ± 1.87	0.05 ± 0.78	0.902 †
∆ Leucocytes, mm^3^	−468.84 ± 1757.39	−297.16 ± 1523.73	−469.98 ± 1501.62	0.844 *
∆ LMR	−1.57 ± 7.29	−0.35 ± 2.44	−0.84 ± 2.26	0.460 †
∆ CRP, mg/dL	−0.90 ± 18.27	−1.04 ± 3.46	0.00 ± 0.313	0.474 †

* ANOVA; † Kruskal–Wallis; EVOO: extra virgin olive oil; DieTBra: traditional Brazilian diet; NLR: neutrophil-to-lymphocyte ratio; LMR: lymphocyte-to-monocyte ratio; CRP: C-reactive protein; SD: standard deviation; CI: confidence interval.

## Data Availability

The data presented in this study are available on request from the corresponding author.

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
