# Peer review of "The Effectiveness of Extra Virgin Olive Oil and the Traditional Brazilian Diet in Reducing the Inflammatory Profile of Individuals with Severe Obesity: A Randomized Clinical Trial"

_nutrients, 2021, doi:10.3390/nu13114139_

Round 1
Reviewer 1 Report
I thank the authors for the effort and commitment in writing the article entitled “The effectiveness of extra virgin olive oil and the traditional Brazilian diet in reducing the inflammatory profile of individuals with severe obesity: a randomized clinical trial”.
This study, which is part of a bigger RCT, known as the DieTBra Trial, investigates the role of two nutritional interventions, Brazilian Diet and Extra-Virgin Olive Oil (EVOO), on the inflammatory profile in severe obesity.
EVOO positive effects have been largely evaluated on many diseases and health issues, such as on sarcopenia, weight loss in obese patients. However, there is no published data on inflammatory markers and nutritional intervention with this product and a dietary pattern in severe obesity. To-date, EVOO, alone and in association with Mediterranean Diet (MedDiet), is known to have ibuprofen-like properties, but no-one has investigated quantitively the reduction of inflammation based on biomarkers, yet. Moreover, less is known about Brazilian Diet and its potential benefits on health. So, this study may represent an innovative point of view on the possible effectiveness of nutritional interventions on weight-related issues.
The introductive chapter is a little bit confused. The first paragraphs focus on obesity co-morbidities and adverse health outcomes but there’s no connection between the sentences. For example: “the mental health […] highly prevalent in such individuals” should be rephrased to get a better link with the whole text. The second part, instead, describes the inflammatory markers that increase in obese people. However, biomarkers are only mentioned but there is no explanation about the reason why it may be useful in the present study. A better explanation could be useful.
The “material and methods” chapter is divided into specific paragraphs. The inclusion and exclusion criteria are well structured, trying to reduce all the possible confounding elements.
There are some critic points in:
- The creation of study groups. This study, as well as the main RCT DieTBra, divides the study population into three groups: the EVOO only-group, the DieTBra only-group and the EVOO + DieTBra group. However, in my opinion, it would have been useful to introduce a fourth “control” group, with no nutritional modifications, to have a baseline reference.
- The choice of inflammatory biomarkers: in the introductive chapter, the writers mentioned many markers associated with inflammation and obesity, such as IL-6 and TNF-α. However, in this study, blood samples were evaluated for: CRP, NLR, LMR and leucocytes count. In my opinion, it would have been useful to also analyse IL-6 and TNF-α in consideration of their strong correlation with adiposity.
- The length of the intervention period: maybe, it would have been more useful to study this population for a longer time, because a window of 12 weeks may not be enough to see significant differences in severely obese people. We have to consider that these participants started from a strong surrounding pro-inflammatory status and they only introduced small nutritional variations.
The “results” chapter is well done: tables are easy to understand and their location in the middle of the text helps the reader have a clear vision of data. However, there’s a point that could be improved. It is about the comparison and the statistically significance of data. Data are analysed comparing baseline levels at time zero and at the end of the intervention period (12 weeks) in the same group. However, there isn’t an evaluation of the possible differences among the three groups. It would have been useful to have a look transversally.
The “discussion and conclusions” chapter explains correctly the limitations and the benefits of this study.
In conclusion, this article has an interesting aim but it lacks in innovation. The interesting point is to analyse whether other dietary patterns may be as beneficial as MedDiet. However, it also lacks in evaluating some important biomarkers and the intervention window is too short. Moreover, as said in the article, there is no blinding: because of nutritional intervention, single or double blinding can’t be performed. So, even if the researchers have introduced some strategies (such as “no contact” among study participants), the risk of bias remains very high.
Eventually, I suggest considering a transverse comparison between the possible differences between the three intervention groups.
Author Response
Dear reviewer 1, please see the attachment.

Reviewer 2 Report
This study, about the effect of virgin olive oil and a traditional Brazilian diet on the inflammation profile of obese individuals, is interesting.
This study is part of a series of publications (at least 5 already published) on the results of a randomized clinical intervention on nutrition.
Since my expertise is not mainly on inflammation, I will let other experts comment on the usefulness of the biological markers selected.
I know that this clinical trial has already been evaluated several times but data on the waist circumference and waist to hip ratio would have been much more informative about the relationship between obesity and related complications. Do the authors have such anthropometric data?
My first and main concern is that the authors claim that the variations in NLR and CRP showed “important clinical relevance despite no statistically significant differences. The authors can mention a “trend” but definitely cannot present on one hand that a variation of one marker is not significant and on the other hand explain that they observe a “clear reduction”. Please correct this whenever possible.
I know that it may be already presented in other publications, but could the authors add the percentage of male/female in table 1, and the bodyweight at baseline and after 12 weeks in table 2.
Indeed, a double-blind study is often difficult to set up in nutrition, but the authors don’t say if the researcher knew which diet participants were receiving before the end of the analysis of the clinical trial. So were the researchers blinded? Still, about blindness in this article, I know that nutritional interventions are difficult to blind. However, could the authors give information about the number of sachets not consumed in each group, or if published elsewhere, could authors contextualize such information with the results presented. It could shed some light on diet preference in each group.
Also, could the authors present data associated with weight loss in the DietBra group and discuss it when it concerns? I believe that it could give important information on the ratios presented since an immune modulation could be associated with weight loss.
In the introduction, the authors mention that other studies have found that a regular intake of olive oil could reduce the level of inflammatory markers. What could be the biological mechanism behind this protective effect of olive oil?
In the discussion, each time it is possible, could the authors add data (numbers) when they describe other studies. For instance, it would be important to know the values of the CRP when the authors mention a significant reduction of CRP after a dietary oleic acid supplementation.
Round 2
Reviewer 1 Report
The revision paper is ok for me
Author Response
Dear reviewer, thank you very much for the positive evaluation of our study.
Reviewer 2 Report
Thank you for the corrections and the answers to the comments and questions.
Sorry but I still have doubt about the statistical significance of the difference of body weight between baseline and 12 weeks, for instance 120.43+/-20.85 versus 118.85+/-19.28 (p<0.001). Is that possible to have such low p value with high standard deviations ? Could the authors check the statistical test used?
